# Fine-Grained Power Gating Using an MRAM-CMOS Non-Volatile Flip-Flop

**DOI:** 10.3390/mi10060411

**Published:** 2019-06-20

**Authors:** Jaeyoung Park, Young Uk Yim

**Affiliations:** Qualcomm CDMA Technologies, Qualcomm Technologies Inc., San Diego, CA 92121, USA; yyim@qti.qualcomm.com

**Keywords:** STT-MRAM, flip-flop, power gating, low-power

## Abstract

An area-efficient non-volatile flip flop (NVFF) is proposed. Two minimum-sized Metal-Oxide-Semiconductor Field-Effect Transistor (MOSFET) and two magnetic tunnel junction (MTJ) devices are added on top of a conventional D flip-flop for temporary storage during the power-down. An area overhead of the temporary storage is minimized by reusing a part of the D flip-flop and an energy overhead is reduced by a current-reuse technique. In addition, two optimization strategies of the use of the proposed NVFF are proposed: (1) A module-based placement in a design phase for minimizing the area overhead; and (2) a dynamic write pulse modulation at runtime for reducing the energy overhead. We evaluated the proposed NVFF circuit using a compact MTJ model targeting an implementation in a 10 nm technology node. Results indicate that area overhead is 6.9% normalized to the conventional flip flop. Compared to the best previously known NVFFs, the proposed circuit succeeded in reducing the area by 4.1× and the energy by 1.5×. The proposed placement strategy of the NVFF shows an improvement of nearly a factor of 2–18 in terms of area and energy, and the pulse duration modulation provides a further energy reduction depending on fault tolerance of programs.

## 1. Introduction

Power gating has been researched as an effective energy-reduction technique [1,2,3]. This reduces static power consumption by shutting power off. However, data needs to be transferred to another storage component before the power-off and restored after the power-on [4,5]. Such transfers of data introduce energy and area overheads. Therefore, it is important to develop a low overhead temporary storage component and an efficient strategy of the use of the storage components. Off-chip memories have been used for the temporary storage [2,3]; however, a complex interface between the off-chip memories and a chip stands in the way of wide adoption of such off-chip memories in power gating scheme.

Embedded non-volatile flip-flops (NVFFs) are promising enablers to fine-grained power gating because these do not require a complex interface to transfer state from/to the external storage. One critical issue is an overhead to store/restore data onto non-volatile temporary storage of the NVFF before/after the power-down. It is universally the case when adding a new feature (e.g., non-volatility) to the existing flip-flop. However, what is the best way to build a low overhead NVFF? A magnetic tunnel junction (MTJ) of a spin torque transfer magnetic memory (STT-MRAM) is a candidate for the non-volatile storage of the NVFF because the MTJ does not occupy the silicon area; the MTJ is placed between metal layers. However, the area and energy overheads can be significant if write and read circuits for the MTJ are not carefully optimized.

In this paper, we propose an area-efficient MRAM-Complementary Metal-Oxide-Semiconductor (CMOS) hybrid non-volatile flip-flop. We reutilize the existing CMOS flip-flop for transferring data to/from MTJs to reduce the area overhead. Only two minimum-sized transistors are added. In addition, the energy overhead for storing data onto MTJs is reduced by 50% by reusing a write current to write two MTJs. We evaluated the proposed NVFF circuit using a compact MTJ model and a 10 nm Predictive Technology Model (PTM) MOSFET model [6,7,8]. The proposed NVFF has an improvement by a factor of 4–23 in terms of the area over state-of-the-art circuits. In addition, energy for the storing operation is reduced by 1.5× compared to the best previously proposed NVFF circuits.

We also propose optimization strategies for the use of the proposed NVFF. Because the proposed NVFF introduces a non-negligible area overhead compared to the conventional D flip-flop (FF), it is important to use the proposed NVFF carefully to minimize the area overhead. Replacing all the conventional FFs in a design with NVFFs imposes a large area penalty. Therefore, we first analyze a circuit and then place the NVFFs only for a selected module which can minimize the area penalty. We place the proposed NVFF in a module that has a low ratio of the flop area to total. Because a module contains flops, combinational logic circuits, and passive devices, the area penalty by the NVFF is minimized where FFs occupy the relatively small area to the total. In the other words, the area increase by the NVFF becomes relatively small if the other components in the same modules occupy the greater portion of the module.

In addition, the write pulse duration for the NVFF needs to be carefully optimized due to the stochastic nature of the MTJ write. The MTJ write process itself is fundamentally stochastic and the actual time to the completion varies dramatically with the distribution having a very long tail [9,10,11]. This means that write energy also varies quite significantly and the write energy can be wasteful if the applied write pulse duration is not carefully selected. Instead of using the conventional deterministic strategy with a fixed pulse duration that guarantee a target error probability, we exploit this stochastic property to save more energy by adjusting the pulse duration adaptively. A key insight is that high fault-tolerance programs can endure more error from the NVFF so that we can reduce the write pulse duration for the programs to save write energy even if the NVFF itself introduces higher error probability.

We demonstrate the optimization strategies on an OpenSPARC design which is an open-source version of UltraSPARC processor [12]. Four programs—matrix multiply, sort, bzip2, and prime—are also selected for this experiment [13]. Analysis indicates the placement shows an improvement of nearly a factor of 2–18 in terms of area and energy and the pulse duration modulation maximizes energy savings of the proposed NVFF for programs have high fault tolerance. The detailed analysis are presented in the following sections.

## 2. MRAM-CMOS Non-Volatile Flip-Flops

### 2.1. State-of-the-Art MRAM-CMOS NVFFs

MRAM-CMOS NVFFs typically need extra circuits for writing and reading MTJs. Multiple realizations of the extra circuits, which use additional write drivers and sense amplifiers, have been proposed [14,15,16,17]. In [15], two NAND gates, seven inverters, and three NMOS switch transistors are used for the external write driver and the sense amplifier with a significant reduction of D-Q delay. In [16], four NOR gates, four inverters, and 16 NMOS transistors are used to reduce C-Q delay and sensing currents.

In [14], only three extra transistors are added for writing and reading MTJs because the existing cross-coupled inverter pair of the conventional D-FFs is used to assist with storing and restoring operations of MTJs. Figure 1 shows the storing and restoring operations of the NVFF [14]. For the storing operation, $MTJ_{A}$ is written to antiparallel (AP) state by lowering Reset-ENable (REN) signal when QS is logical ’H’ for Q = 1. $MTJ_{B}$ is written to parallel (P) state by raising the REN and Set-ENable (SEN) signals in the second write phase. The restoring operation is achieved by the regenerative feedback of the inverter pair because a different voltage is developed between QS and QSb nodes when $MTJ_{A}$ and $MTJ_{B}$ have different resistances. This is why the NVFF requires only three additional transistors. However, sizable transistors are needed to drive sufficient current with low Vgs because the Vgs is dropped by IR drop through an MTJ (Vgs = Vdd − I × $R_{\mathrm{MTJ}}$). Moreover, the storing energy is doubled because two MTJs need to be written in different phases.

### 2.2. Current Reutilization NVFF

We propose a current reutilization technique to reduce energy and area overheads. The main idea that a single write current for an MTJ can be used to write another MTJ. Instead of applying two separate current pulses to write two MTJs at different phases, we can write two MTJs using a single write current at the same time. The current reutilization should not introduce large area overhead. We achieved this by inserting one minimum-sized NMOS transistor because two MTJs can be placed on the same current path via the NMOS transistor (M1) as shown in Figure 2. A write current is passed through $MTJ_{A}$, M1, and $MTJ_{B}$ when a switch transistor (M1) is turned on for Q = 1. The $MTJ_{A}$ is written to the AP state because the current direction is from the pinned layer (PL) to the free layer (FL) of the MTJ, and $MTJ_{B}$ becomes the P state because the current direction is reversed (FL→PL). For storing Q = 0, a write current goes through $MTJ_{B}$, M1, and $MTJ_{A}$; therefore, the situation is reversed ($MTJ_{B}$ = AP, $MTJ_{A}$ = P). The proposed current reutilization technique allows for writing both MTJs using one write current at the same time. Thus, we could reduce the write current by 50%, resulting in a half storing energy. In contrast to an NVFF of Yamamoto et al. [14], an inverter pair drives a write current and a minimum-sized NMOS transistor is only used as a switch. In addition, a full Vdd is applied to a gate of the inverter pair during the storing operation.

The restoring operation is achieved by another minimum-sized transistor (M2). This reutilizes the inverter pair of the existing D-FF. Different voltages are developed between QS and QSb of the slave latch by two MTJs that have different resistances when M1 and M2 are turned on after the power-up.

### 2.3. Evaluation of the Proposed NVFF

We designed the proposed NVFF using a 10 nm predictive technology model (PTM) MOSFET model and a compact MTJ model [7,8]. Key parameters of the perpendicular MTJ is described in Table 1. Figure 3 shows Simulation Program with Integrated Circuit Emphasis (SPICE) simulation results of the proposed NVFF using the models. The proposed NVFF operates as a conventional D-FF in normal operations. On top of the D-FF, non-volatile operations are added. The storing operation is performed before the power-down. The output Q is stored in MTJs when SEN is raised. $MTJ_{A}$ is written to the AP state and $MTJ_{B}$ is the P state for Q = 1. During the power-down mode, the output Q is lowered. The Q is restored when power is up again at 28.6 ns (restoring operation).

We compare the proposed NVFF with the state-of-the-art NVFF circuits as shown in Table 2. The proposed NVFF shows an improvement of nearly a factor of 2–17 in terms of restoring energy compared to the state-of-the-art NVFF circuits. Note that the restoring energy is reduced by 50% if MTJ and CMOS devices are the same as an NVFF [14]. The proposed NVFF implemented in an advance technology node and the greater part of the storing energy reduction comes from the technology scaling.

The relative area increase is only 6.9% (2/29) because only two minimum-sized transistors are added to the conventional D flip-flop (FF) that has 29 transistors. We did not directly compare the area because technologies of the reference circuits are different, and the actual area strongly depends on the layout optimization. Thus, we used a relative area overhead to the D-FF of each technology for this comparison. Note that the size of the PMOS transistor in the inverter is assumed to be 2× NMOS transistor. The relative area overheads of state-of-the-art NVFF architectures are from 28.0% to 160.0%. Therefore, the proposed NVFF has an improvement of nearly a factor of 4–23 in terms of the area overhead compared to state-of-the-art NVFF circuit.

A simulated delay of the restoring operation is 10 ps and a storing time is set to 6.6 ns to have a sufficiently low error probability. We computed the error probability of the proposed NVFF. We used the following probability model derived in [11] using a Neel–Brown relaxation formula to compute error probability. The model describes the switching probability $P_{SW}\left(t,I\right)$, which is the probability of switching occurring for a pulse duration *t* at current *I*:(1)$$P_{SW}\left(t,I\right)=1-e^{-\frac{t}{\tau_{0}e^{\Delta (1-I/I_{c0})}}},$$
where Δ is the thermal stability factor and $\tau_{0}$ is is the inverse of the thermal attempt frequency that has a typical value of 1 ns [10,11]. $I_{C0}$ is a critical current and *I* is an applied current to write. A computed write error probability is $1.5\times 10^{-13}$ where an average write current is 24.6μA and a storing time is 6.6 ns.

## 3. Optimization Strategies for the Proposed NVFF

In this section, we describe optimization strategies of the use of the proposed NVFF. Because the proposed NVFF introduces a non-negligible area overhead compared to the conventional D-FF, it is important to use the proposed NVFF carefully to minimize the area overhead. In addition, the MTJ write process itself is fundamentally stochastic and the actual time to completion varies dramatically with the distribution having a very long tail [9,10,11]. This means that write energy also varies quite significantly and the write energy can be wasted if the applied pulse duration is not carefully adjusted. We propose a two-phase optimization strategy: (1) a static NVFF placement in a design phase and (2) dynamic pulse width modification at runtime. The proposed two-phase flow is illustrated in Figure 4. In a design phase, we place the NVFF only in a module that is able to maximize the benefit of the NVFF. At runtime, we dynamically adjust the write pulse duration to save more energy for a program that has high fault tolerance. The details are described as follows.

### 3.1. Pre-Fabrication Optimization: A Module-Based Placement

A key question for the optimization is where the NVFF is placed to reduce static power while minimizing the area overhead. Replacing all the conventional FFs with NVFFs imposes a large area penalty. Therefore, we first analyze a circuit and then place the NVFFs only for a selected module. This is a fine-grained (or cluster-based) power gating approach. We characterize a circuit using two metrics, static power and a ratio of the FF area to the total, and then place the NVFFs in a module that has high static power and low area ratio. Because the area penalty is minimized where FFs occupy the relatively small area to the total. In addition, more static power can be saved if the module itself consumes high static power. Because the power gating can reduce static (leakage) power by shutting off power supply and it has no impact on dynamic power, placing the NVFFs in a high-static-power module can save more static power. Because a module generally contains not only FFs but also has combinational logic circuits and passive devices, the power gating can also reduce the static power of the combinational logic circuits and passive devices in the same module too.

We demonstrate the proposed optimization strategy in designing OpenSPARC T1 core, which is an open-source version of UltraSPARC processor. We first synthesized all modules and performed the placement and routing using *Synopsys* 32 nm EDK standard cell library [23]. We used *Synopsys Design Compiler, IC Compiler, and Primetime* for synthesizing, placement and routing, and static timing and power analysis, respectively [24,25,26]. We then selected seven high computational modules, and analyzed area and static power. As shown in Table 3, ALU (exu_alu) and decoder (ifu_dec) modules have fewer FFs than the other five modules. This results in lower area ratio to total. The increased area is less than 1% if the conventional FFs in the modules are replaced with the proposed NVFFs. Among two modules, the static power of the ALU is higher than that of the decoder. Therefore, we selected the ALU for a module to place the proposed NVFF. The placement shows an improvement of nearly a factor of 2–18 in terms of area and energy compared to the other modules.

All performances of seven modules are summarized in Table 3. The area and power are computed using *Synopsys Primetime*. Storing and restoring energy from the 10 nm PTM model are scaled up based on a constant field scaling method [27] because 32 nm standard cell library is used for the placement and routing of the OpenSPARC core. A break-even time ($T_{\mathrm{breakeven}}$) is determined when energy saving by the power-gating is equal to the energy overhead by storing and restoring operations.

### 3.2. Post-Fabrication Optimization: A Pulse Width Modulation

We now describe a post-fabrication optimization strategy. The main idea is that a write pulse duration can be adaptively adjusted to reduce the write energy overhead for programs which have high fault tolerance. Because of a trade-off between energy and error probability of the propose NVFF, the write energy can be reduced by sacrificing error probability. This is true where each program has its unique fault tolerance even if the hardware design remains unchanged. In other words, some programs can tolerate more error so that we can use a shorter pulse duration for the programs to save more energy.

To implement this idea, we first examine the fault tolerance of programs to validate whether the fault tolerance varies over programs. Four programs—matrix multiply, sort, bzip2, and prime—are selected for this experiment. Gate-level simulations of the programs are performed on an OpenSPARC core using *Synopsys VCS* to inject faults and monitor the final outputs [28]. The fault injection process is based on a gate-level simulation that is halted at a randomly-determined cycle. The gate-level simulator extracts outputs of the combinational blocks and flip-flops for the cycle of interest. We inject faults (e.g., flipped value) on the flops based on the probabilities of their occurrence. After the injection, the analysis continues to the end of programs to determine whether the fault has been masked or a system failure has occurred. Outcomes from the fault injection are compared to a golden fault-free run. System failures by the fault injection are categorized as one of the following: detected unrecoverable error (DUE), Output match, silent data corruption (SDC), Hung, or Masked. We did this fault injection process for four programs. As shown in Figure 5, the most frequent category is Masked (above 90% of all cases). The second highest category is DUE, followed by SDC and Output match. The Hung case is not observed in the simulation. Among the four programs, bzip2 shows the lowest system failure rate, 1.2%. The DUE is only 0.6%, whereas the other programs are above 2.2%. This clearly shows that bzip2 has better tolerance in this experiment; therefore, a shorter pulse duration can be used for the program to save more energy.

We also examine how much energy we can save by adjusting the pulse duration. Figure 6 shows the error rate and the expected energy of a flop at different pulse duration. The error probability is inversely proportional to the write pulse duration as Equation (Equation 1), and the expected energy is linearly increased by the pulse duration while the error rate is exponentially decreased. At 6.6 ns, a write error probability is $1.5\times 10^{-13}$ and energy for a storing and restoring operations is 0.2 pJ. For short pulses such as 3.3 ns, the computed error probability is increased to $3.9\times 10^{-7}$ while the energy consumption is reduced by half. Because of such trade-off, the applied pulse duration for each program needs to be carefully selected based on the target error probability of the system even if the pulse duration modulation strategy maximizes energy savings of the proposed NVFF.

In order to control the pulse duration, a control circuit is necessary. However, the area overhead per flop would be negligible because one circuit can control all FFs in a chip. In addition, the pulse duration is selected at software-level because the program information is needed.

## 4. Conclusions

In this paper, a novel area efficient NVFF is proposed. The relative area overhead is 6.9%, and the proposed NVFF shows an improvement of nearly a factor of 4–23 in terms of area overhead compared to state-of-the-art NVFF designs. The write current for the restoring operation is reduced by 50% using the proposed current-reuse technique. To our knowledge, the proposed NVFF enables a fine-grained power gating without significant area overhead. Compared to the best previously known NVFFs, the proposed NVFF succeeds in reducing the area by 4.1× and the energy by 1.5×.

Two optimization strategies for reducing area and energy overheads are also proposed: NVFF placement and pulse duration modulation strategies. We demonstrated the placement strategy on an OpenSPARC T1 core design. Analysis indicates that the placement on the ALU shows an improvement of nearly a factor of 2–18 in terms of area and energy compared to the other modules. We also demonstrated the fault tolerance variation over programs and the adaptive pulse duration strategy for the energy savings.

## Figures and Tables

**Figure 1 micromachines-10-00411-f001:**
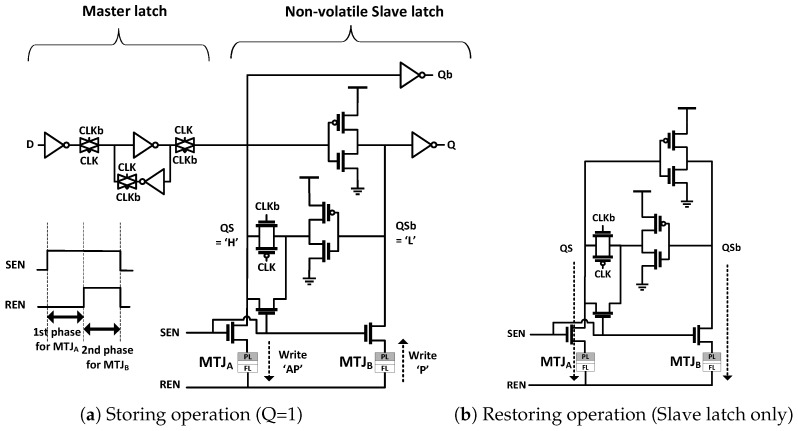
Schematic of an non-volatile flip flop (NVFF) of Yamamoto et al. [14].

**Figure 2 micromachines-10-00411-f002:**
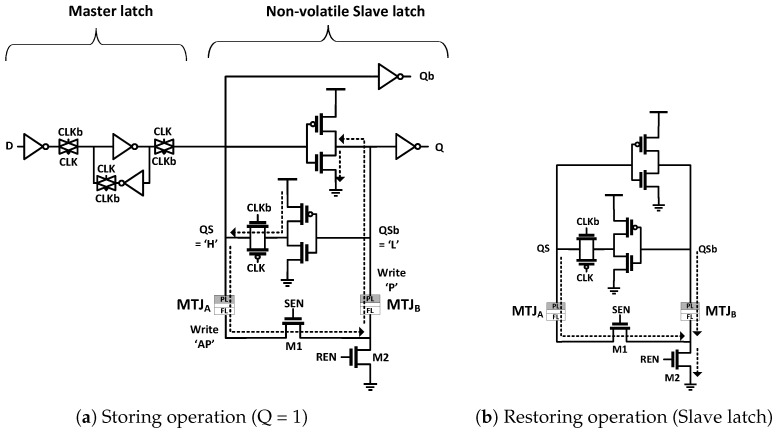
Schematic of the proposed current reutilization NVFF.

**Figure 3 micromachines-10-00411-f003:**
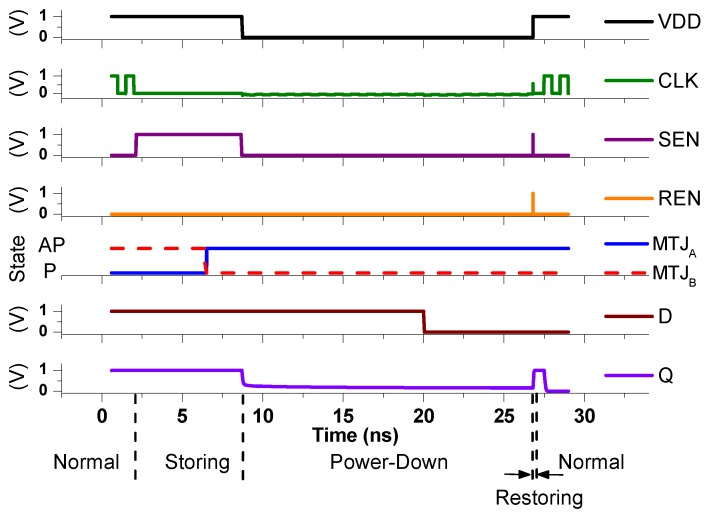
Waveforms of each node of the proposed NVFF. (Output ‘1’ is stored and restored. *x*-axis denotes time and the *y*-axes indicates voltage (V) or states of MTJs.)

**Figure 4 micromachines-10-00411-f004:**
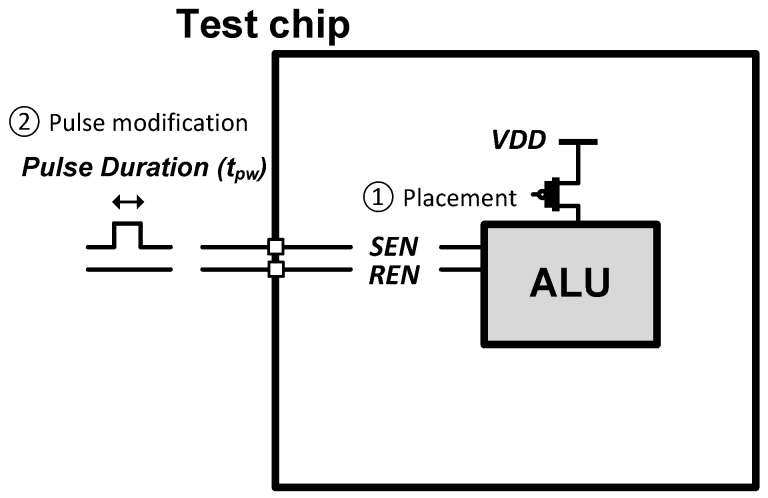
Overview of the proposed two-phase optimization flow.

**Figure 5 micromachines-10-00411-f005:**
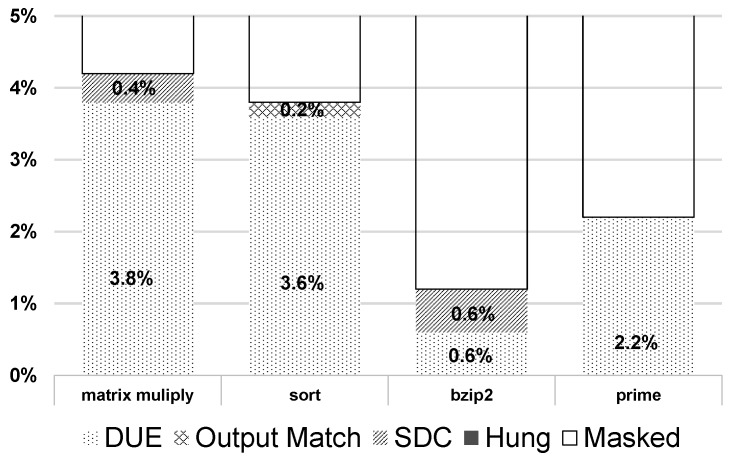
System failure results by fault injection.

**Figure 6 micromachines-10-00411-f006:**
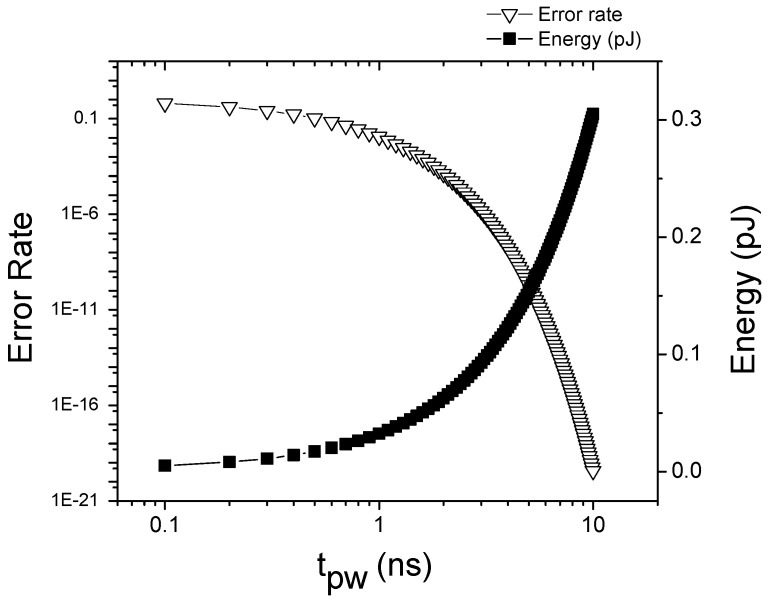
Error rate and the expected energy of the proposed NVFF at different write pulse duration.

**Table 1 micromachines-10-00411-t001:** Key parameters of perpendicular magnetic tunnel junction (MTJ) [6,7,8].

Parameter	Value	Unit
Intrinsic critical current	24	μA
Thermal stability factor	58	
Tunnel Magnetoresistance ratio (TMR)	∼100	%
Diameter of MTJ	20	nm
Out-of-plane magnetic field	0.4	T

**Table 2 micromachines-10-00411-t002:** Performance summary and comparison with the state-of-the-art NVFFs.

		MRAM-Based							FeRAM-	ReRAM-
		**This Work**	[17]	[16]	[15]	[18]	[14]	**[19] ^a^**	[20]	[21]
Technology node (nm)		10	90	45	45	90	65	65	130	65
Area overhead ^b^ (%)		6.9	131.0	160.0	120.0	103.4	109.0	28.0	64.0	32.0
Energy (pJ)	Storing	0.2	175.5	1.9	1.6	0.3	5.0	0.5	2.4	-
	Restoring	0.002	-	0.171	0.007	-	0.349	0.197	-	-
Delay (ns)	Storing	6.6	-	-	-	10.0	29.5	6.4	1640.0	-
	Restoring	0.01	0.169 ^c^	2.0	0.184	1.0	2.0	2.0	1230.0	16.0
C-Q delay (ps)		43.8	318.1 ^c^	68.8	186.2	67.2	73.8	-	-	<1 ns
Power-Delay Product (fJ)		0.3	2.8 ^c^	1.1	2.3	0.7	1.4	-	-	-

^a^ Spin Hall Effect MTJ, ^b^ normalized to the conventional D flip-flop, ^c^ Data from [22]. FeRAM-Ferroelectric RAM, ReRAM-Resistive RAM.

**Table 3 micromachines-10-00411-t003:** Performance summary of seven modules.

	FF Area (μm${}^{2}$)	Total Area (μm${}^{2}$)	FF/Total (%)	NVFF Area (μM${}^{2}$)	Increased Area (%)	$P_{\mathrm{static}}$ (mW)	$E_{\mathrm{storing}}$ + $E_{\mathrm{restoring}}$ (pJ)	$T_{\mathrm{breakeven}}$ (ns)
exu_alu	429.5	15,022.5	2.9	459.1	0.2	1.8	72.6	40.8
exu_div	3714.6	12,218.2	30.4	3970.9	2.1	0.2	628.0	3924.0
exu_ecl	2319.3	6869.5	33.8	2479.4	2.3	0.1	392.1	4292.9
exu_rml	1729.2	4340.0	39.8	1848.5	2.7	0.4	733.1	1929.6
ifu_dec	277.5	4049.1	6.8	296.7	0.5	0.4	46.9	119.7
ifu_fcl	1785.6	5991.8	29.8	1908.8	2.1	0.5	301.9	616.6
ffu_dp	5466.6	13,722.1	39.8	5843.8	2.7	1.3	924.2	716.1

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
