# Peer review of "Fine-Grained Power Gating Using an MRAM-CMOS Non-Volatile Flip-Flop"

_micromachines, 2019, doi:10.3390/mi10060411_

Round 1
Reviewer 1 Report
This paper discusses the use of magnetic tunnel junction nonvolatile memories for non-volatile flip-flop implementation, that has for aim a reduction in energy via power gating. The authors introduce three novel techniques in this paper: a) Current reuse which allows to store two opposite phases with the same write current, b) reduction in area overhead at design phase through adding only 2 NMOS transistors to a D-Flip-Flop and using optimized placement techniques, and c) dynamic modulation to reduce energy per operation.
The paper is clearly written, with a wealth of work on both conceptual design and simulation for predicted performance. Given that the predictions indicate the design could be a competitive one in overhead reduction using NVFFs, I would recommend accepting the paper pending some minor revisions:
a) The authors use several acronyms that are not defined and could impede the readability for someone not fully in the field: SEN/REN, AP/P etc... It will be helpful to define the acronyms that may not be obvious for the reader.
b) In the table comparing current work to other NVFF implementations (page 4, table 2), the authors only focus on MTJ-based devices, although other emerging technologies have been demonstrated recently, such as using FeRAMs and ReRAMs. It would be crucial to add those technologies especially of MTJ-based storage is highlighted in this paper as a leading technology.
c) Figure 3 has no axis labels, making it a little difficult to interpret.
d) On page 4, before-last paragraph, a comparison is made between the area overhead on the proposed device vs other devices, and makes the argument that an absolute area comparison is difficult due to different implementation techniques. However, it will be helpful if the authors could provide a rough indication if they expect this technology to require a larger baseline area due to the arrangement of the transistors (or smaller/comparable one).
e) The pulse-width modulation section is interesting, especially that a compromise can be achieved between bit error rate and energy. However, my concern is that adaptive pulse widths could add hardware overhead on top of the NVFF itself. Moreover, where would the decision on the pulse width be made ?
If those points are adressed, I feel the paper is suitable for publication in MDPI micromachines.
Author Response
We are grateful to the editor and reviewers for their insightful comments and suggestions. This document is our response to the issues raised by the reviewers, listed in a point-by-point fashion. We sincerely hope that we have addressed all of the reviewers’ concerns.
<Response to Reviewer 1’s comments>
General comment: This paper discusses the use of magnetic tunnel junction nonvolatile memories for non-volatile flip-flop implementation, that has for aim a reduction in energy via power gating. The authors introduce three novel techniques in this paper: a) Current reuse which allows to store two opposite phases with the same write current, b) reduction in area overhead at design phase through adding only 2 NMOS transistors to a D-Flip-Flop and using optimized placement techniques, and c) dynamic modulation to reduce energy per operation.
The paper is clearly written, with a wealth of work on both conceptual design and simulation for predicted performance. Given that the predictions indicate the design could be a competitive one in overhead reduction using NVFFs, I would recommend accepting the paper pending some minor revisions:
Comment 1: The authors use several acronyms that are not defined and could impede the readability for someone not fully in the field: SEN/REN, AP/P etc... It will be helpful to define the acronyms that may not be obvious for the reader.
Response 1: Thank you for your comment. We have added the explanation of the acronyms which are marked in red.
Comment 2: In the table comparing current work to other NVFF implementations (page 4, table 2), the authors only focus on MTJ-based devices, although other emerging technologies have been demonstrated recently, such as using FeRAMs and ReRAMs. It would be crucial to add those technologies especially of MTJ-based storage is highlighted in this paper as a leading technology.
Response 2: Thank you for pointing us this. We have added FeRAM- and ReRAM-based NVFFs on Table 2.
Comment 3: Figure 3 has no axis labels, making it a little difficult to interpret.
Response 3: We think you pointed out Y axis because X axis already has a label, Time (ns). We initially had the same concern and tried to add labels on the Y axis. However, unfortunately, there is no room to add any other labels except for the current labels such as “(V)” and “State”. We have added notes for the labels in the title of Figure 3 as below:
“X axis denotes time and Y axes indicate voltage (V) and states of MTJs”
Comment 4: On page 4, before-last paragraph, a comparison is made between the area overhead on the proposed device vs other devices, and makes the argument that an absolute area comparison is difficult due to different implementation techniques. However, it will be helpful if the authors could provide a rough indication if they expect this technology to require a larger baseline area due to the arrangement of the transistors (or smaller/comparable one).
Response 4: We use the size of the conventional D-FF on each technology for the area comparison. This is because an NVFF in an advanced technology always is smaller than NVFFs in older technologies due to a technology scaling. Furthermore, many reference papers do not show an actual FF area, but only describe the number of additional MOSFETs to achieve the non-volatile property. This is why we used an indirect comparison.
Comment 5: The pulse-width modulation section is interesting, especially that a compromise can be achieved between bit error rate and energy. However, my concern is that adaptive pulse widths could add hardware overhead on top of the NVFF itself. Moreover, where would the decision on the pulse width be made?
Response 5: Thank you for this feedback and allowing us to improve our paper. This is true that a control circuit is necessary. However, one circuit can control all FFs in a chip, which means the area overhead per flop would be negligible. Because the pulse duration is selected based on program information, the decision will be made at software-level. In addressing the point, we have added a comment as below in Section 3.2:
“In order to control the pulse duration, a control circuit is necessary. However, the area overhead per flop would be negligible because one circuit can control all FFs in a chip. Also, the pulse duration is selected at software-level because the program information is needed.”

Reviewer 2 Report
This manuscript proposes a new design of non-volatile flip-flop (NVFF) that reduce area and energy cost compared to the state-of-the-art NVFF. It is written well but in lack of some necessary details. Detailed comments are as follows: In section 1, I don’t understand the following sentence in paragraph 4 “We statically place the proposed NVFF in a module that has high static power and a low ratio of the flip flop (FF) area to the total to maximize the static power saving while reducing area overhead”. Why do the authors specifically targeting at high-static-power modules only? Ans how is the static power saving related to the module static power? Similarly, I don’t think “reducing area overhead” only works for low ratio of the FF area, correct? Compared to the state-of-the-art NVFF designs, the proposed one should reduce the area overhead regardless of the FF area ratio. In the rest of this paragraph, it is not clear to me what is the relationship between faults and write pulse duration. I would suggest the authors reorganize this paragraph to make things more clear. About the area cost of the NVFF designs, comparing Fig 1 (state-of-the-art in [11]) and 2 (the proposed one), it seems to me that the proposed design only saves 1 transistor compared to the state-of-the-art design. Looking at table 2, the area overhead of [11] is around 15x of the proposed one’s. Where does the other area reduction come from? From the manuscript, it is not clear to me why the proposed design could achieve such a low area reduction compared to [11]. As for the energy cost of the NVFF designs, I have two questions: Since the authors use a single current to write both MTJs, I assume the voltage needs to be higher? If yes, how higher does the required voltage need to be? If not, why? The authors mentioned in section 2.2, paragraph 1 that the storing energy could be reduced by 50% due to half the write current compared to [11]. In Table 2, it shows much more savings compared to [11]. Could the authors elaborate why? In short, there are some discrepancies between the design description and the results. The authors need to provide more details or explanations to make things more clear for the readers.
Author Response
We are grateful to the editor and reviewers for their insightful comments and suggestions. This document is our response to the issues raised by the reviewers, listed in a point-by-point fashion. We sincerely hope that we have addressed all of the reviewers’ concerns.
<Response to Reviewer 2’s comments>
General comment: This manuscript proposes a new design of non-volatile flip-flop (NVFF) that reduce area and energy cost compared to the state-of-the-art NVFF. It is written well but in lack of some necessary details. Detailed comments are as follows:
Comment 1: In section 1, I don’t understand the following sentence in paragraph 4 “We statically place the proposed NVFF in a module that has high static power and a low ratio of the flip flop (FF) area to the total to maximize the static power saving while reducing area overhead”. Why do the authors specifically targeting at high-static-power modules only? Ans how is the static power saving related to the module static power? Similarly, I don’t think “reducing area overhead” only works for low ratio of the FF area, correct? Compared to the state-of-the-art NVFF designs, the proposed one should reduce the area overhead regardless of the FF area ratio. In the rest of this paragraph, it is not clear to me what is the relationship between faults and write pulse duration. I would suggest the authors reorganize this paragraph to make things more clear.
Response 1: Thank you for your comment. Unfortunately, our original description was unclear. To clarify the point, we have reorganized the paragraph and have added the detailed explanation in Section 1 and 3 (marked in red).
(1) Area overhead – The proposed NVFF is smaller than the other NVFFs; however, the proposed NVFF introduces a non-negligible area overhead compared to the conventional D-FF. That is why we tried to minimize the area penalty by the optimization of the use. We have added a comment as below in Section 1:
“Because the proposed NVFF introduces a non-negligible area overhead compared to the conventional D flip-flop (FF), it is important to use the proposed NVFF carefully to minimize the area overhead. Replacing all the conventional FFs in a design with NVFFs imposes a large area penalty. Therefore, we first analyze a circuit and then place the NVFFs only for a selected module which minimizes the area penalty. We place the proposed NVFF in a module that has a low ratio of the flop area to total. Because a module contains flops, combinational logic circuits, and passive devices, the area penalty by the NVFF is minimized where FFs occupy the relatively small area to the total. In the other words, the area increase by the NVFF becomes relatively small if the other components in the same modules occupy the greater portion of the module.”
(2) Static power – The power gating technique can reduce static (leakage) power by shutting off power supply and it has no impact on dynamic power. Also, a module contains flops, combinational logic circuits, and passive devices. Therefore, the power gating can reduce the static power of the combinational logic and passive devices too by shutting off power of a module. That is why we targeted a high-static-power module to maximize the benefit of the power gating. We have added a comment as below in Section 3.2:
“Because the power gating can reduce static (leakage) power by shutting off power supply and it has no impact on dynamic power, placing the NVFFs in a high-static-power module can save more static power. Because a module generally contains not only FFs but also has combinational logic circuits and passive devices, the power gating can also reduce the static power of the combinational logic and passive devices in the same module too.”
(3) The relationship between faults and write pulse duration – The error probability is inversely proportional to the write pulse duration as Equation (1). This is already explained in Section 2, but we did not emphasize in Section 3 which explains the optimization of the pulse duration. We have added a comment as below in Section 3 and the detailed explanation for the stochastic nature of the MTJ write is Section 1:
“The error probability is inversely proportional to the write pulse duration as Equation (1), and the expected energy is linearly increased by the pulse duration while the error rate is exponentially decreased.”
Comment 2: About the area cost of the NVFF designs, comparing Fig 1 (state-of-the-art in [11]) and 2 (the proposed one), it seems to me that the proposed design only saves 1 transistor compared to the state-of-the-art design. Looking at table 2, the area overhead of [11] is around 15x of the proposed one’s. Where does the other area reduction come from? From the manuscript, it is not clear to me why the proposed design could achieve such a low area reduction compared to [11].
Response 2: We thank the reviewer for this feedback. An NVFF in [11] requires only 3 additional transistors, which is one more transistor compared to the proposed NVFF. However, sizable transistors are needed to drive sufficient current with low Vgs because the Vgs is dropped by IR drop through an MTJ (Vgs = Vdd - I_RMTJ) while the proposed NVFF uses minimum-sized transistors. In contrast to an NVFF of Yammamoto et al., an inverter pair drives a write current and a minimum-sized NMOS transistor is only used as a switch in the proposed NVFF. Also, a full Vdd is applied to a gate of the inverter pair during the storing operation. This is the why we could use minimum-sized transistors while the reference design is needed sizable transistors as described in the draft.
Comment 3: As for the energy cost of the NVFF designs, I have two questions:
Since the authors use a single current to write both MTJs, I assume the voltage needs to be higher? If yes, how higher does the required voltage need to be? If not, why?
The authors mentioned in section 2.2, paragraph 1 that the storing energy could be reduced by 50% due to half the write current compared to [11]. In Table 2, it shows much more savings compared to [11]. Could the authors elaborate why?
Response 3: Because the proposed NVFF implemented in an advance technology node (10nm FINFET), the nominal Vdd is sufficient to drive a critical current of the MTJ. No boosted Vdd is required. Also, your point regarding the storing energy is correct. We only save 50% storing energy if MTJ and CMOS devices are the same. The greater part of the storing energy reduction
comes from the technology scaling.
We actually mentioned 50% energy saving in Section 2.2 – “Thus, we could reduce the write current by 50%, resulting in a half storing energy.”
However, unfortunately, it might not be clear. We have added a comment below in Section III:
“Note that the restoring energy is reduced by 50% if the MTJ and CMOS devices are the same with an NVFF Yammamoto et al. [14]. The proposed NVFF implemented in an advance technology node and the greater part of the storing energy reduction comes from the technology scaling.”

Round 2
Reviewer 1 Report
The authors have addressed my comments in a satisfactory manner, I believe it is acceptable in its current form.
Reviewer 2 Report
I thank the authors for their clear and detailed explanation to my questions. I have no further questions now and this manuscript is good for publication.